# Species of the *Colletotrichum* spp., the Causal Agents of Leaf Spot on European Hornbeam (*Carpinus betulus*)

**DOI:** 10.3390/jof9040489

**Published:** 2023-04-19

**Authors:** Yu-Hang Qiao, Chen-Ning Zhang, Min Li, Huan Li, Yun-Fei Mao, Feng-Mao Chen

**Affiliations:** 1College of Forestry, Nanjing Forestry University, Nanjing 210037, China; 2Collaborative Innovation Center of Sustainable Forestry in Southern China, Nanjing 210037, China; 3Suzhou Forestry Station, Suzhou 215100, China

**Keywords:** *Carpinus betulus*, *Colletotrichum*, leaf spot, identification

## Abstract

European hornbeam (*Carpinus betulus* L.) is widely planted in landscaping. In October 2021 and August 2022, leaf spot was observed on *C. betulus* in Xuzhou, Jiangsu Province, China. To identify the causal agent of anthracnose disease on *C. betulus*, 23 isolates were obtained from the symptomatic leaves. Based on ITS sequences and colony morphology, these isolates were divided into four *Colletotrichum* groups. Koch’s postulates of four *Colletotrichum* species showed similar symptoms observed in the field. Combining the morphological characteristics and multi-gene phylogenetic analysis of the concatenated sequences of the internal transcribed spacer (ITS) gene, Apn2-Mat1-2 intergenic spacer (ApMat) gene, the calmodulin (CAL) gene, glyceraldehyde3-phosphate dehydrogenase (GAPDH) gene, Glutamine synthetase (GS) gene, and beta-tubulin 2 (TUB2) genes, the four *Colletotrichum* groups were identified as *C. gloeosporioides*, *C. fructicola*, *C. aenigma*, and *C. siamense*. This study is the first report of four *Colletotrichum* species causing leaf spot on European hornbeam in China, and it provides clear pathogen information for the further evaluation of the disease control strategies.

## 1. Introduction

European hornbeam (*Carpinus betulus* Linnaeus) belongs to the family Betulaceae and is mainly distributed in temperate and subtropical regions. It is native to peripheral forests around Europe, Asia Minor, and the Caspian Sea, and is often mixed with oak and beech. It can grow well above 1000 m above sea level [1]. In Iran, European hornbeam is the main tree species in the wood industry, with excellent technical performance and great application potential. It is mainly used for manufacturing tool handles, furniture, and paper, and it is also an excellent wood for railway sleeper production and dam reinforcement after preservative treatment [2]. European hornbeam is also very popular in urban green spaces and parks, and it has excellent characteristics of cold resistance, drought resistance, and pruning resistance. European hornbeam has been selected as an important tree species in garden construction since the Italian Renaissance [3]. In addition, it has been reported that many anticancer substances can be extracted from the young stems and leaves of European hornbeam, such as pheophorbide A (PHA) and some triterpenoids [4,5]. Therefore, European hornbeam has great research value and practical potential.

The genus *Colletotrichum* Corda is the only genus of Glomerellaceae [6] and one of the ten most important plant pathogenic fungi in the world [7]. The fungi of the genus *Colletotrichum* are distributed worldwide, with diverse host plants, including more than 3000 species of monocot and dicot plants [8,9]. Some *Colletotrichum* species can also cause human infection and inflammation, such as *Colletotrichum gloeosporioides* [10,11].

Before the 1990s, the classification of *Colletotrichum* was mainly based on morphological characteristics. The morphological classification of the genus *Colletotrichum* is mainly based on the morphology and size of conidia, the appressorium, the sporulation structure and conidiophores, and the presence and morphology of chlamydospores, setae, and sclerotia [12,13,14]. However, because of the instability of these characteristics, the classification of *Colletotrichum* is very confusing [12,15,16]. With the rapid development of molecular biology, the method based on morphology combined with molecular biological identification has gradually been used for the classification of *Colletotrichum* and has been widely used [17,18]. The internal transcribed spacer (ITS) is easy to analyze; however, the sequence of ITS fragments within species is relatively consistent. Therefore, it is difficult to accurately identify a species by its ITS fragments [19]. Multi-gene sequence analysis is increasingly applied to the classification of *Colletotrichum*, and the frequently used gene loci include the calmodulin (CAL) gene, glyceraldehyde3-phosphate dehydrogenase (GAPDH) gene, beta-tubulin 2 (TUB2) gene, chitin synthase (CHS-1) gene, actin (ACT) gene, Glutamine synthetase (GS) gene, Apn2-Mat1-2 intergenic spacer (ApMat) gene, etc.

*Colletotrichum fructicola*, *C. aenigma*, *C. gloeosporioides*, and *C. siamense* all belong to the *Colletotrichum gloeosporioides* species complex. These four species are more or less typical of the generalized *C. gloeosporioides* reported in the past half century in terms of morphology [20]. *C. fructicola* was first reported on *Coffea* in Thailand, and its hosts are diverse in terms of biology and geography, including *Coffea*, *Dalbergia hupeana*, and *Millettia speciosa* in different countries [21,22,23]. *C. aenigma* was named based on its mysterious biological and geographical distribution. It was first discovered in Italy and Japan, and consistent with the prediction of Weir et al., it has been gradually reported in China, Thailand, and South Korea in recent years [24,25,26]. *C. gloeosporioides* is common worldwide and easily found on *Citrus*, but it also infects other hosts, including papaya and *Rubia cordifolia* [27,28,29]. *C. siamense* was also first reported on *Coffea* in Thailand, and its hosts also exhibit diversity in biology and geography, such as *Plukenetia volubilis* in China, *Annona muricata* in Brazil, and *Capsicum annuum* in the Andaman and Nicobar Islands [30,31,32].

At present, the diseases reported for European hornbeams mainly include the powdery mildew of leaves, the canker of branches, and root rot. The pathogens of powdery mildew vary among areas. Piątek reported that *Oidium carpini* caused powdery mildew in Poland [33], Vajna reported that the pathogen of powdery mildew in Hungary was *Erysiphe carpinicola* [34], and Pastircakova found that the new pathogen of powdery mildew in Slovakia was *Erysiphe arcuata*, which Chinan also reported in Romania [35,36]. There are few reports of canker and root rot. Rocchi reported that branch cankers in Italy were caused by *Anthostoma decipiens* [3]. Mao reported that *Fusarium oxysporum* caused root rot in Jiangsu Province, China [37]. Recently, in Xuzhou, Jiangsu Province, China, European hornbeam was found to have symptoms of leaf spots, which affected the local landscape and economic development in this area.

This study aims to identify the pathogenic fungi that cause leaf spot disease from the aspects of phylogeny and morphology and to study their biological characteristics and pathogenicity, ultimately to provide a theoretical basis for the prevention and control of leaf spot disease in European hornbeam.

## 2. Materials and Methods

### 2.1. Sampling and Fungal Isolation

The field survey was investigated in Xuzhou Urban Garden Company (34.28° N, 118.03° E) in October 2021 and August 2022 in Xuzhou, Jiangsu Province. Xuzhou has a temperate monsoon climate with four distinct seasons, the annual sunshine hours are 2284–2495 h, the sunshine rate is 52–57%, the average annual precipitation is 800–930 mm, and the rainy season precipitation accounts for 56% of the whole year.

Diseased leaves were collected from a 1–2 m part of European hornbeam. Approximately 30 diseased samples were collected from 10 European hornbeam trees which were scattered in the field. Fungi isolation was conducted on the second day after field survey. The diseased leaves were disinfected in 1% sodium hypochlorite for 90 s, rinsed in sterile water twice for 30 s, and dried with sterile paper. Then, the tissues from the margin of the lesions (0.2 cm × 0.2 cm) were excised, incubated on 2% potato dextrose agar (PDA) supplemented with 100 mg/L ampicillin sodium, and incubated in the dark at 25 °C for 4 days. Fungal hyphae grown from the leaf tissues were picked up and transferred to fresh PDA within 2–4 days [38].

### 2.2. Pathogenicity Tests

Healthy European hornbeam saplings with a height of approximately 1 m were obtained from Xuancheng Garden Greening Co., Ltd. in Xuancheng, Anhui Province.

Before the pathogenicity experiment, the surfaces of the leaves were sprayed with 75% alcohol 2–3 times, and then the above operation was repeated with sterile water to remove the residual alcohol; then, they were dried with absorbent paper, or we waited for the surfaces to dry. The spore suspension (10^6^ conidia·mL^−1^) was sprayed 2–3 mL onto the leaves using a 10 mL plastic sprinkling can, and hornbeam leaves were treated with sterile water as the control.

Each of the treatment and control groups contained five leaves, and each treatment consisted of one seedling. All of the seedlings under different treatments were kept in a 25 °C greenhouse with high humidity under natural light conditions, and the development of symptoms was observed daily. The experiments were conducted twice.

To complete Koch’s postulates, as previously mentioned, the fungus was reisolated from the margin tissue of the diseased lesions that developed from the inoculated tissue and were identified via molecular and phylogenetic analysis.

### 2.3. Morphological Characteristics

Fresh mycelium blocks were cut from the edge of three-day-old colonies and transferred to fresh PDA medium. After 4 days of incubation in the dark at 25 °C, the colony morphology was observed and recorded.

To observe the morphology of conidia, fresh mycelium pieces were cut off and transferred to fresh potato dextrose broth (PDB) supplemented with 100 mg/L ampicillin sodium. Then, the PDB bottles containing the mycelium pieces were placed on a shaking table and shaken at a rotating speed of 200 rpm in the dark at a temperature of 25 °C. After 2 days, the culture solution was collected and filtered with sterile filter cloths to collect the conidia. Appressoria were induced via cultivation on the surface of a hydrophobic coverslip [39]. Asci or ascospores were obtained from the ascomata that grew for 2–3 weeks on PDA or SNA in darkness at 25 °C. Then, each structure was observed to generate 30 measurements using a ZEISS Axio Imager A2m microscope (ZEISS), and the size of each structure was measured using the cross-assay method [40].

### 2.4. Phylogenetic Analysis

Fungal hyphae were collected from fresh colonies using sterilized scalpels. Genomic DNA was extracted using a CTAB Extraction Solution Kit (Leagene Biotechnology, Beijing, China). Then, all of the DNA extracts were stored at −20 °C for subsequent use.

Six nuclear gene regions were amplified and sequenced, including the ITS, CAL, GAPDH, TUB2, ACT, and CHS-1 regions. The primers and PCR conditions are shown in Table 1. Amplification was performed in an Eppendorf Nexus Thermal Cycler (Eppendorf) in a volume of 50 μL, which consisted of 4 μL of genomic DNA, 2 μL of forward/reverse primer (0.01 nmol/μL), 25 μL of 2× Green Taq Mix (Vazyme, Nanjing), and 17 μL of double-distilled H_2_O. PCR products were sequenced by Sangon Biotech Co., Ltd. (Nanjing, China).

The sequences were analyzed using MAFFT [49] in PhyloSuite v. 1.2.2 [50] and manually trimmed to ensure maximum sequence similarity.

Maximum likelihood (ML) analysis and Bayesian inference (BI) analysis were used to mutually corroborate the phylogenetic reconstructions. IQ-TREE v. 1.6.8 [51] was used for inferring the ML phylogenies under the edge-linked partition model for 100,000 ultrafast bootstraps. MrBayes v. 3.2.6 [52] was used for inferring BI phylogenies, and the initial quarter of the sampled data was discarded as burn-in. ModelFinder [53] was used to select the best-fit model on the basis of the Akaike information criterion (AIC). According to the AIC, the best-fitting model for ML analysis was GTR + F + I + G4, with 1,000,000 ultrafast [54] bootstrap replicates determining the branch stability, while the model for BI analysis was GTR + F + I + G4 under 2 parallel runs of 1,000,000 generations. The phylogenetic tree was viewed by FigTree v. 1.4.4.

## 3. Results

### 3.1. Field Survey and Symptoms in the Field

The field survey was investigated in Xuzhou Urban Garden Company (34.28° N, 118.03° E) in October 2021 and August 2022 in Xuzhou, Jiangsu Province. There were about 5000 European hornbeam in the field, approximately 20% of the European hornbeam showed symptoms of leaf spots, and diseased leaves accounted for approximately 15–20% of the diseased European hornbeam.

Most of the spots were distributed along the edge of the European hornbeam leaves, and the spot wounds tended to expand inward. In addition, some serious disease spots caused leaf shape loss or leaf curling. The spots were brown to dark brown, some areas of the lesion appeared to be grayish-white, and the margin of a part of the lesions appeared as a pale green halo (Figure 1).

### 3.2. Fungal Isolation

A total of 23 fungal strains were isolated from the diseased leaf samples of European hornbeam. Based on ITS sequences, 23 strains belonged to the genus *Colletotrichum*. According to the density of hyphae and the distribution of pigment on the reverse side of colonies, 23 *Colletotrichum* strains were divided into 4 groups, with quantities of 5 (group 1), 12 (group 2), 4 (group 3), and 2 (group 4).

### 3.3. Morphological Characteristics

One representative isolate was selected from each *Colletotrichum* group for further study (XZEC11 from group 1, XZEC21 from group 2, XZEC31 from group 3, and XZEC41 from group 4).

The colonies of XZEC11 isolates produced white aerial hyphae with loose marginal hyphae, and the back of the colonies was light orange-red (Figure 2A,B). The colonies of XZEC21 had fluffy aerial hyphae with loose marginal hyphae, and both sides were all white. The center of the reverse side appeared to be irregular and slightly grayish-green (Figure 3A,B). The aerial hyphae of XZEC31 were compact and raised in the center, and the reverse side was pale orange (Figure 4A,B). The colonies of XZEC41 exhibited fluffy aerial hyphae with loose marginal hyphae, and both sides were all white. The center of the front side was gray, and the reverse side showed blackish-green annular concentric rings (Figure 5A,B).

The colonies of XZEC11, XZEC21, and XZEC41 on SNA produced white, sparse aerial hyphae, and the center area of the colonies of XZEC31 was slightly dense compared with the marginal hyphae; the colonies of XZEC31 produced sparser aerial hyphae than those of the other three *Colletotrichum* groups (Figure 2C,D, Figure 3C,D, Figure 4C,D and Figure 5C,D).

The conidia of the four *Colletotrichum* groups were obtained after shaking cultivation with a rotating speed of 200 rpm in the dark at a temperature of 25 °C. Generally, the structures of the four groups appeared to be cylindrical, straight, and hyaline, and they were all aseptate. Additionally, the conidia of XZEC11 were blunt and rounded at both ends, and the longitudinal middle was slightly concave (Figure 2E–H). The conidia of XZEC21 were thinner than those of the other groups, and one end of the conidia was slightly convex (Figure 3E–H). The conidia of XZEC31 showed a slightly standard semicircle at both ends, and one end was convex (Figure 4E–H). The conidia of XZEC41 had a slightly sharp end and were slightly concave in the middle (Figure 5E–H). The size of each group is shown in Table 2.

The appressoria of the four *Colletotrichum* groups were induced via cultivation on the surface of the hydrophobic coverslip in darkness at 25 °C for 12 h. The appressoria were all olive green. The shape ranged from nearly round to nearly oval, and irregular shapes were observed. Most of the conidia of the four *Colletotrichum* groups extended from one end to form appressoria, and a few conidia could extend from both ends to form an appressorium (Figure 2I–L, Figure 3I–L, Figure 4I–L and Figure 5I–L). The sizes of the appressoria of the four *Colletotrichum* groups were similar (Table 2).

The ascomata developed on the surface of the colony or under the mycelium, and sterile blades were used to pick the ascomata out of the colony and cut them into pieces to obtain the asci and ascospores. The ascomata of XZEC11 were irregular, and the ascomata produced in the medium were black, while those produced on the surface of the medium were brown (Figure 2M–P). The ascospores of XZEC11 were aseptate, spindle-shaped, slightly curved, and with round ends (Figure 2Q–T); the ascospores of XZEC21 were hyaline, one-celled, and aseptate (Figure 3N). The asci of XZEC21 were clavate, thin-walled, and eight-spored (Figure 3M).

### 3.4. Pathogenicity Tests

For each *Colletotrichum* group, one representative isolate was selected for the pathogenicity test (XZEC11 from group 1, XZEC21 from group 2, XZEC31 from group 3, and XZEC41 from group 4). Four isolates of *Colletotrichum* were pathogenic, and the inoculated European hornbeam leaves showed lesions similar to the previous symptoms that were observed naturally; nevertheless, the controls remained healthy 10 days after inoculation. Most of the lesions occurred at the edge of the leaves, and a few occurred in some central areas of the leaves (Figure 6). According to the appearance of the lesions, lesions caused by XZEC1 and XZEC4 were scattered, and their area was small. Lesions caused by XZEC31 were mainly distributed along the edge of leaves with a long and narrow shape, and some infected areas of leaves were missing. Lesions caused by XZEC21 were mainly distributed along the edge of leaves and were wider than those of XZEC31.

### 3.5. Phylogenetic Analysis

Eleven representative *Colletotrichum* strains (three strains of group 1, three strains of group 2, three strains of group 3, and two strains of group 4) were selected for phylogenetic analysis on the basis of the sequences of the six nuclear gene regions. The sequences of the 11 *Colletotrichum* isolates were deposited in GenBank (Table 3). The sequences of the 6 fragments of these 11 *Colletotrichum* isolates were concatenated, and the concatenated matrix consisted of 2837 nucleotide characteristics, viz., ACT: 1–265, ApMat: 266–998, CAL: 999–1650, CHS: 1651–1901, GAPDH: 1902–2156, GS: 2157–3072, ITS: 3073–3618, and TUB: 3619–4349. The sequences of 51 strains of the genus *Colletotrichum* were used to construct a phylogenetic tree, with *Colletotrichum hippeastri* (CBS 241.78) included as the outgroup. The GenBank accession numbers of the 51 sequences of *Colletotrichum* are shown in Table 3.

ML and BI analyses produced similar topologies, providing statistical support for the evolutionary relationships of fungal isolates, and a consensus tree with clade support from bootstrap proportions (BPs) and Bayesian posterior probabilities (BPPs) was generated (Figure 7). As shown in Figure 7, 11 *Colletotrichum* strains were placed in 4 different clades with high support values: 3 strains of group 1 were clustered with *C. gloeosporioides* (BP/BPP = 100%/1), 3 strains of group 2 were clustered with *C. fructicola* (BP/BPP = 98%/0.98), 3 strains of group 3 were clustered with *C. aenigma* (BP/BPP = 99%/1), and 2 strains of group 4 were clustered with *C. siamense* (BP/BPP = 99%/1).

## 4. Discussion

A graceful appearance with strong phenotypic plasticity and excellent technical properties of timber gives European hornbeams an important role in urban landscaping and economy. However, leaf spot deteriorates the leaf appearance and affects apical dominance, reducing the quality of wood [55]. In this study, *C. gloeosporioides*, *C. fructicola*, *C. aenigma*, and *C. siamense* were identified as the causal agents of leaf spot on European hornbeam.

Generally speaking, the morphological structure will be identified initially to determine the genus of *Colletotrichum*. In early studies, most of the identification of the *Colletotrichum* species was based on the shape of the conidia [56,57]. The size of the conidia of *C. gloeosporioides* in this study was similar to that reported by Huang et al. [58], Kim et al. [59], and Chen et al. [60], but larger than that reported by Chen et al. [61]. The conidia of *C. siamense* were similar to those reported by Kim et al. [59] and Cao et al. [62], but smaller than those reported by Zhang et al. [63]. The conidia of *C. aenigma* were similar to those reported by Zheng et al. [64] but larger than those reported by Wang et al. [65]. The conidia of *C. fructicola* were similar to those reported by Cai et al. [66] and Huang et al. [58] but shorter than those reported by Costa et al. [67] and Zheng et al. [64]. Other structures, such as appressoria, also exhibit various degrees of difference. This phenomenon in which the sizes of the same structure are not similar could be because of different growth conditions or a loss or change under repeated subculturing [20], similar to the results reported for the asci and ascospores of *C. siamense* and *C. aenigma,* which we failed to induce in this study. Significantly, we isolated *C. aenigma* from *Acer rubrum* in 2020 in the same nursery in Xuzhou [68], and the method of inducing asci and ascospores was developed during the cultivation of *C. aenigma* (2020). Except for asci and ascospores, the colonies of the two *C. aenigma* species were not quite the same (Figure 8). The colonies of *C. aenigma* (2020) were relatively flat, with a relatively fluffy texture. The middle area of *C. aenigma* colonies (2022) was raised, the height dropped gradually from the middle to the edge, and the texture was relatively tight.

During the cultivating of these two *Colletotrichum aenigma*, we used the same PDA medium with the same formula, cultivated them in the same incubator in the darkness at 25 °C, and the positions in the incubator were also very close. However, despite this, the colony morphology and the ability to produce asci and ascospores changed. It is confusing that the morphology of the same *Colletotrichum* species changed just because of the different hosts. Therefore, it is not accurate to identify the *Colletotrichum* species only from morphology, even if it is a 100% identical species. So, more accurate identification methods are needed to distinguish the *Colletotrichum* species, such as multi-gene-combined phylogenetic analysis.

According to the previous literature, ACT, CHS-1, GAPDH, HIS3, ITS, and TUB2 could be used to classify the majority of the *Colletotrichum* species [6], and three additional loci (ApMat, CAL, and GS) have been used for the *C. gloeosporioides* species complex [20,69]. Five conventional genes (ACT, CHS-1, GAPDH, ITS, and TUB2), four specific genes (ApMat, CAL, and GS), and one additional specific gene (APN2) were used in this study, and four species in this study were separated from the *C. gloeosporioides* species complex. The combined phylogenetic tree was consistent with trees presented in other studies [61,70,71,72].

Due to its strong environmental adaptability, the *Colletotrichum* species can cause leaf spot and fruit diseases with huge losses in agricultural and forestry production worldwide [21,65,73,74,75,76,77,78,79,80,81,82]. Furthermore, changes in the climate, human activities, and other factors may cause fungi host jumping within the plants in the nursery [83]. It is very likely that *C. aenigma*, which we isolated in 2020 and 2022, has experienced this, and maybe *C. aenigma* and the other three species also jumped to the other hosts (except for *Acer rubrum* and European hornbeams), though we have not found this yet. Therefore, more reports about local leaf spot diseases caused by *Colletotrichum* species may be produced in the future.

## 5. Conclusions

In conclusion, this study identified *C. gloeosporioides*, *C. fructicola*, *C. aenigma*, and *C. siamense* as the pathogens causing leaf blight on European hornbeam, posing a new and emerging threat to European hornbeam. This research represents the first detailed study of the pathogenicity, morphology, and phylogeny of four *Colletotrichum* species on European hornbeam in China. Further research exploring the infection cycle of this emerging disease in European hornbeam remains to be conducted, and strategies for the control of this new pathological system should be identified.

## Figures and Tables

**Figure 1 jof-09-00489-f001:**
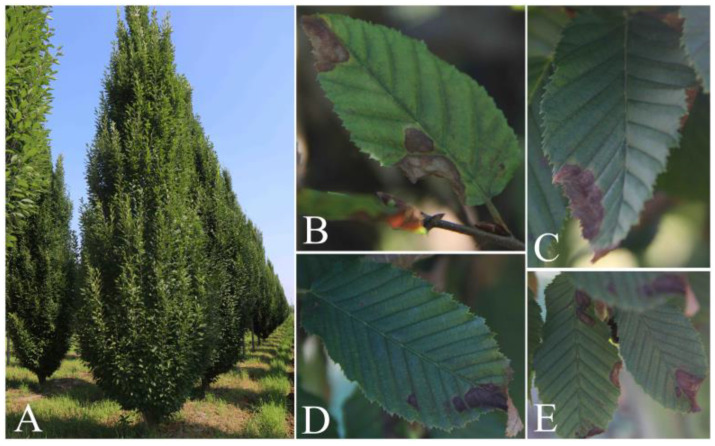
European hornbeam leaves with disease symptoms under natural conditions. (**A**) Diseased tree in the field. (**B**–**E**) Diseased leaves of European hornbeam.

**Figure 2 jof-09-00489-f002:**
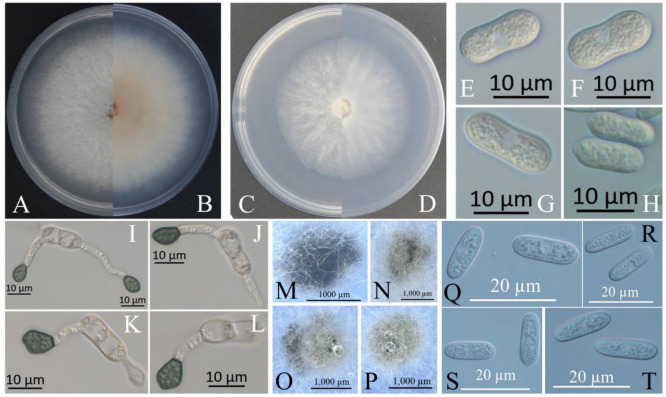
Morphological characteristics of XZEC11. (**A**,**B**) Front and reverse views of 4-day-old fungus on PDA, respectively. (**C**,**D**) Front and reverse view of 4-day-old fungus on SNA, respectively. (**E**–**H**) Conidia. (**I**–**L**) Appressoria. (**M**–**P**) Ascomata developed in or on SNA after cultivation for 2–3 weeks. (**Q**–**T**) Ascospores. Scale bars: (**E**–**L**) = 10 μm; (**M**–**P**) = 1000 μm; (**Q**–**T**) = 20 μm.

**Figure 3 jof-09-00489-f003:**
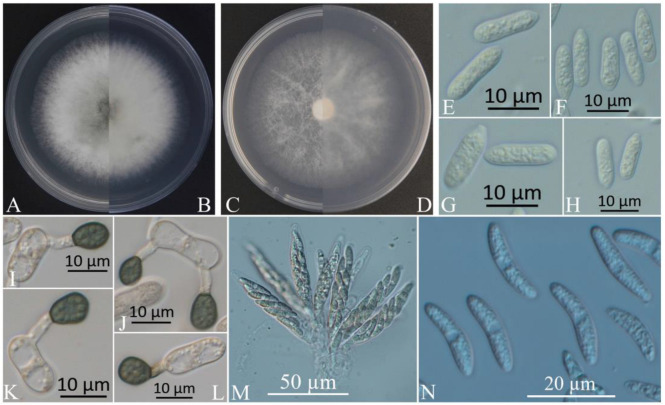
Morphological characteristics of XZEC21. (**A**,**B**) Front and reverse views of 4-day-old fungus on PDA, respectively. (**C**,**D**) Front and reverse views of 4-day-old fungus on SNA, respectively. (**E**–**H**) Conidia. (**I**–**L**) Appressoria. (**M**) Asci. (**N**) Ascospores. Scale bars: (**E**–**L**) = 10 μm; (**M**) = 50 μm; (**N**) = 20 μm.

**Figure 4 jof-09-00489-f004:**
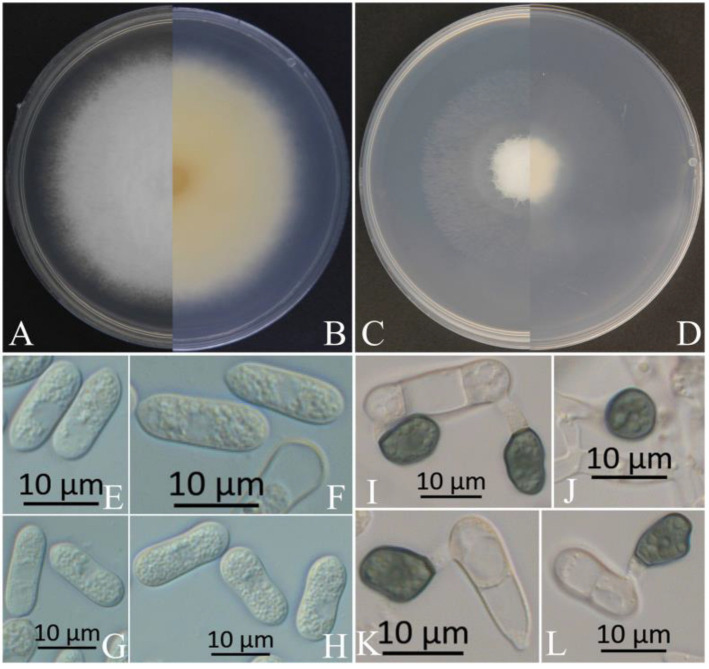
Morphological characteristics of XZEC31. (**A**,**B**) Front and reverse views of 4-day-old fungus on PDA, respectively. (**C**,**D**) Front and reverse views of 4-day-old fungus on SNA, respectively. (**E**–**H**) Conidia. (**I**–**L**) Appressoria. Scale bars: (**E**–**L**) = 10 μm.

**Figure 5 jof-09-00489-f005:**
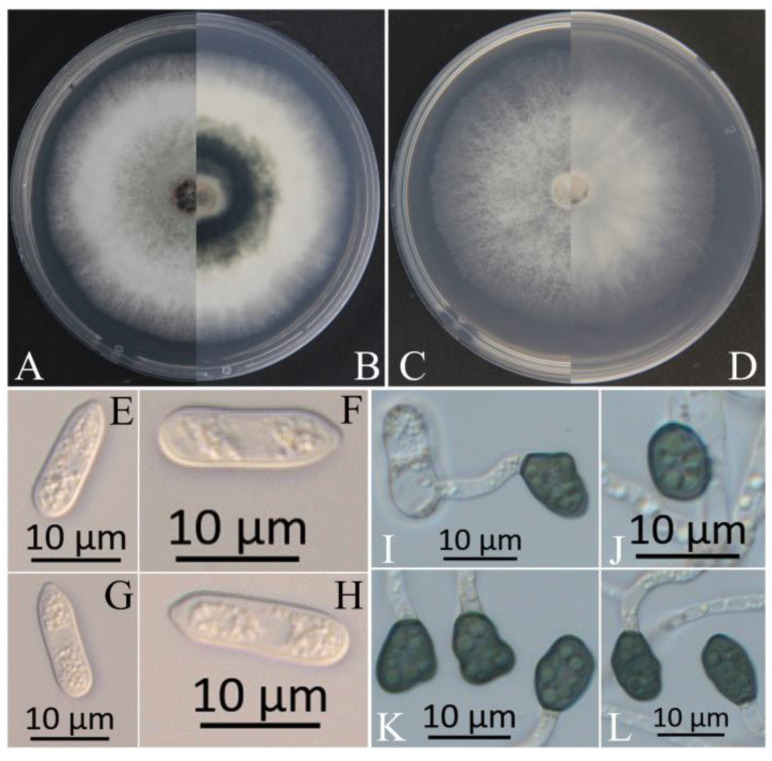
Morphological characteristics of XZEC41. (**A**,**B**) Front and reverse views of 4-day-old fungus on PDA, respectively. (**C**,**D**) Front and reverse views of 4-day-old fungus on SNA, respectively. (**E**–**H**) Conidia. (**I**–**L**) Appressoria. Scale bars: (**E**–**L**) = 10 μm.

**Figure 6 jof-09-00489-f006:**
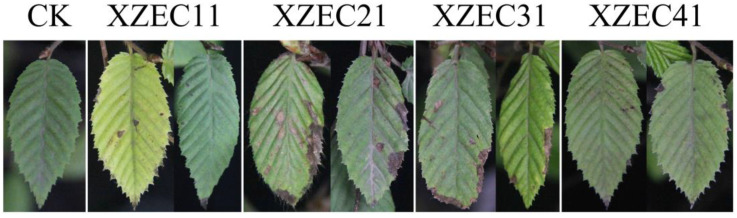
Symptoms on European hornbeam leaves 10 days after inoculation with spore suspensions (10^6^ conidia/mL) of XZEC11, XZEC21, XZEC31, and XZEC41.

**Figure 7 jof-09-00489-f007:**
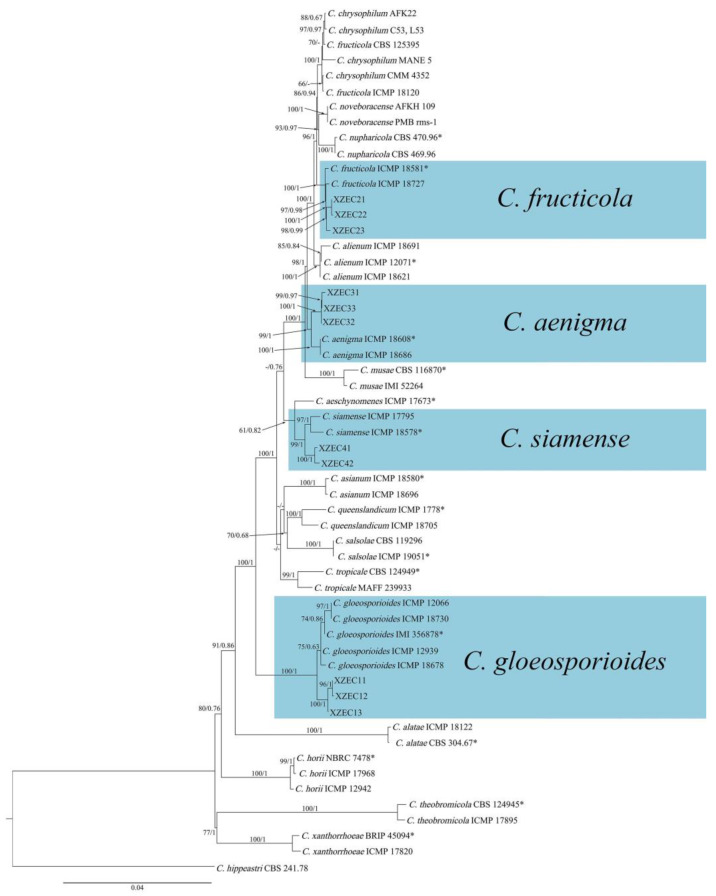
Phylogenetic tree generated with the concatenated sequences of the ITS, ACT, ApMat, CAL, CHS−1, GAPDH, GS, and TUB2 genes using maximum likelihood analysis. The tree generated by Bayesian inference had a similar topology. Bootstrap support values above 60% (before the slash marks) and Bayesian posterior probabilities above 0.60 (after the slash marks) are given at each node (BP/BPP). *Colletotrichum hippeastri* (CBS 241.78) was used as an outgroup. Ex-type strains are marked with (*).

**Figure 8 jof-09-00489-f008:**
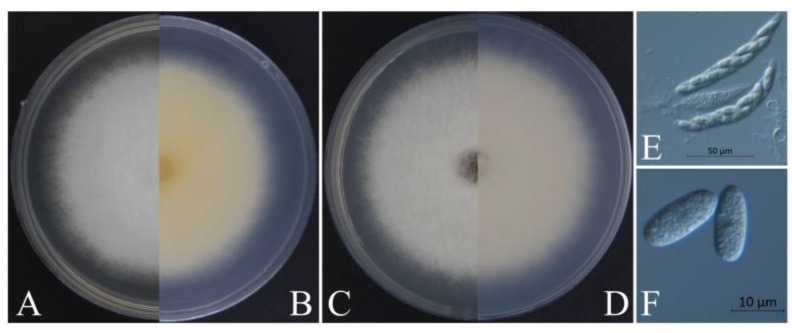
Morphological characteristics of *Colletotrichum aenigma.* (**A**,**B**) Front and reverse views of 4-day-old fungus on PDA of *C. aenigma* (2022), respectively. (**C**,**D**) Front and reverse views of 4-day-old fungus on SNA of *C. aenigma* (2020), respectively. (**E**) Asci of *C. aenigma* (2020), scale bar = 50 μm. (**F**) Ascospores of *C. aenigma* (2020), scale bar = 10 μm.

**Table 1 jof-09-00489-t001:** Primers used in this study, with sequences, PCR conditions, and references.

Gene	Primer	Sequence (5′-3′)	PCR Conditions	References
Internal transcribed spacer (ITS)	ITS1	CTTGGTCATTTAGAGGAAGTAA	Denaturation for 4 min at 95 °C, followed by 35 cycles of 30 s at 95 °C, 30 s at 52 °C, and 45 s at 72 °C, with a final extension of 7 min at 72 °C	[41,42]
	ITS4	TCCTCCGCTTATTGATATGC		
Calmodulin (CAL)	CL1C	GAATTCAAGGAGGCCTTCTC	Denaturation for 4 min at 95 °C, followed by 35 cycles of 30 s at 95 °C, 30 s at 59 °C, and 45 s at 72 °C, with a final extension of 7 min at 72 °C	[20]
	CL2C	CTTCTGCATCATGAGCTGGAC		
Glyceraldehyde-3-phosphate dehydrogenase (GAPDH)	GDF	GCCGTCAACGACCCCTTCATTGA	Denaturation for 4 min at 95 °C, followed by 35 cycles of 30 s at 95 °C, 30 s at 60 °C, and 45 s at 72 °C, with a final extension of 7 min at 72 °C	[43]
	GDR	GGGTGGAGTCGTACTTGAGCATGT		
β-tubulin (TUB2)	T1	AACATGCGTGAGATTGTAAGT	Denaturation for 4 min at 95 °C, followed by 35 cycles of 30 s at 95 °C, 30 s at 55 °C, and 45 s at 72 °C, with a final extension of 7 min at 72 °C	[44,45]
	Bt2b	ACCCTCAGTGTAGTGACCCTTGGC		
Actin (ACT)	ACT-512F	ATGTGCAAGGCCGGTTTCGC	Denaturation for 4 min at 95 °C, followed by 35 cycles of 30 s at 95 °C, 30 s at 58 °C, and 45 s at 72 °C, with a final extension of 7 min at 72 °C	[46]
	ACT-783R	TACGAGTCCTTCTGGCCCAT		
Chitin synthase 1 (CHS-1)	CHS-79F	TGGGGCAAGGATGCTTGGAAGAAG	Denaturation for 4 min at 95 °C, followed by 35 cycles of 30 s at 95 °C, 30 s at 58 °C, and 45 s at 72 °C, with a final extension of 7 min at 72 °C	[46]
	CHS-345R	TGGAAGAACCATCTGTGAGAGTTG		
Glutamine synthetase (GS)	GSF	ATGGCCGAGTACATCTGG	Denaturation for 4 min at 95 °C, followed by 35 cycles of 30 s at 95 °C, 30 s at 59 °C, and 45 s at 72 °C, with a final extension of 7 min at 72 °C	[47]
	GSR	GAACCGTCGAAGTTCCAC		
Apn2- Mat1-2 intergenic spacer (ApMat)	CgDL-F6	AGTGGAGGTGCGGGACGTT	Denaturation for 4 min at 95 °C, followed by 35 cycles of 30 s at 95 °C, 30 s at 58 °C, and 45 s at 72 °C, with a final extension of 7 min at 72 °C	[48]
	CgMAT1F2	TGATGTATCCCGACTACCG		

**Table 2 jof-09-00489-t002:** The size of morphological structures of four *Colletotrichum* groups.

Groups/Isolates	Conidia	Appressoria	Ascospore
XZEC11	14.0–18.3 × 5.5–7.8	6.8–12.1 × 5.4–8.1	14.1–16.9 × 4.4–6.0
XZEC21	11.1–14.8 × 3.6–5.7	7.1–11.0 × 5.7–8.4	19.0–21.8 × 3.7–4.9
XZEC31	12.9–17.5 × 5.4–7.3	7.0–12.5 × 4.8–8.6	——
XZEC41	12.2–18.6 × 3.8–5.7	6.3–12.3 × 5.0–8.3	——

The description of the size is length (μm) × width (μm). The number of each structure observed is 30.

**Table 3 jof-09-00489-t003:** Sequences of *Colletotrichum* species used in this study.

Species, (Culture/Isolate Number ^a^)	Country, Host	GenBank Accession Number ^b^							
		ITS	CAL	GAPDH	TUB2	ACT	CHS-1	GS	ApMat
*C. aenigma* (**ICMP 18608**)	Israel, *Persea americana*	JX010244	JX009683	JX010044	JX010389	JX009443	JX009774	JX010078	KM360143
*C. aenigma* (ICMP 18686)	Japan, *Pyrus pyrifolia*	JX010243	JX009684	JX009913	JX010390	JX009519	JX009789	JX010079	—
*C. aenigma* (XZEC31 ^c^)	China, *Carpinus betulus*	OQ352860	OQ427912	OQ427918	OQ427921	OQ427909	OQ427915	OQ695544	OQ695541
*C. aenigma* (XZEC32)	China, *C. betulus*	OQ352861	OQ427913	OQ427919	OQ427922	OQ427910	OQ427916	OQ695545	OQ695542
*C. aenigma* (XZEC33)	China, *C. betulus*	OQ352862	OQ427914	OQ427920	OQ427923	OQ427911	OQ427917	OQ695546	OQ695543
*C. aeschynomenes* (**ICMP 17673**)	USA, *Aeschynomene virginica*	JX010176	JX009721	JX009930	JX010392	JX009483	JX009799	JX010081	—
*C. alatae* (**CBS 304.67**)	India, *Dioscorea alata*	JX010190	JX009738	JX009990	JX010383	JX009471	JX009837	JX010065	KC888932
*C. alatae* (ICMP 18122)	Nigeria, *Dioscorea alata*	JX010191	JX009739	JX010011	JX010449	JX009470	JX009846	JX010136	—
*C. alienum* (ICMP 18691)	Australia, *Persea americana*	JX010217	JX009664	JX010018	JX010385	JX009580	JX009754	JX010074	—
*C. alienum* (**ICMP 12071**)	New Zealand, *Malus domestica*	JX010251	JX009654	JX010028	JX010411	JX009572	JX009882	JX010101	KM360144
*C. alienum* (ICMP 18621)	New Zealand, *Persea americana*	JX010246	JX009657	JX009959	JX010386	JX009552	JX009755	JX010075	—
*C. asianum* (ICMP 18696)	Australia, *Mangifera indica*	JX010192	JX009723	JX009915	JX010384	JX009576	JX009753	JX010073	—
*C. asianum* (**ICMP 18580**)	Thailand, *Coffea arabica*	FJ972612	FJ917506	JX010053	JX010406	JX009584	JX009867	JX010096	FR718814
*C. chrysophilum* (AFK22)	USA, Apple/Honeycrisp	MN625456	MN622857	MN632505	MN622866	—	—	MN622843	MN622875
*C. chrysophilum* (C53, L53)	Uruguay, Apple/Galaxy	MZ562285	MZ562249	MG491675	MG491716	—	—	MZ562258	MZ562276
*C. chrysophilum* (CMM 4352)	Brazil, Musa sp.	KX094254	KX094064	KX094184	KX094286	—	—	KX094205	KX094326
*C. chrysophilum* (MANE 5)	Brazil, Apple/Gala	KT806271	MZ595288	KT806294	KT806281	—	—	MZ595299	MZ595266
*C. fructicola* (ICMP 18120)	Nigeria, *Dioscorea alata*	JX010182	JX009670	JX010041	JX010401	JX009436	JX009844	JX010091	—
*C. fructicola* (CBS 125395)	Panama, *Theobroma cacao*	JX010172	JX009666	JX009992	JX010408	JX009543	JX009873	JX010098	—
*C. fructicola* (**ICMP 18581**)	Thailand, *Coffea arabica*	JX010165	FJ917508	JX010033	JX010405	FJ907426	JX009866	JX010095	JQ807838
*C. fructicola* (ICMP 18727)	USA, *Fragaria × ananassa*	JX010179	JX009682	JX010035	JX010394	JX009565	JX009812	JX010083	—
*C. fructicola* (XZEC21 ^c^)	China, *C. betulus*	OQ352857	OQ427882	OQ427888	OQ427891	OQ427879	OQ427885	OQ695538	OQ695535
*C. fructicola* (XZEC22)	China, *C. betulus*	OQ352858	OQ427883	OQ427889	OQ427892	OQ427880	OQ427886	OQ695539	OQ695536
*C. fructicola* (XZEC23)	China, *C. betulus*	OQ352859	OQ427884	OQ427890	OQ427893	OQ427881	OQ427887	OQ695540	OQ695537
*C. gloeosporioides* (**IMI 356878**)	Italy, *Citrus sinensis*	JX010152	JX009731	JX010056	JX010445	JX009531	JX009818	JX010085	JQ807843
*C. gloeosporioides* (ICMP 12939)	New Zealand, *Citrus* sp.	JX010149	JX009728	JX009931	—	JX009462	JX009747	—	—
*C. gloeosporioides* (ICMP 12066)	New Zealand, *Ficus* sp.	JX010158	JX009734	JX009955	—	JX009550	JX009888	—	—
*C. gloeosporioides* (ICMP 18730)	New Zealand, *Citrus* sp.	JX010157	JX009737	JX009981	—	JX009548	JX009861	—	—
*C. gloeosporioides* (ICMP 18678)	USA, *Pueraria lobata*	JX010150	JX009733	JX010013	—	JX009502	JX009790	—	—
*C. gloeosporioides* (XZEC11 ^c^)	China, *C. betulus*	OQ352863	OQ427897	OQ427903	OQ427906	OQ427894	OQ427900	OQ695532	OQ695529
*C. gloeosporioides* (XZEC12)	China, *C. betulus*	OQ352864	OQ427898	OQ427904	OQ427907	OQ427895	OQ427901	OQ695533	OQ695530
*C. gloeosporioides* (XZEC13)	China, *C. betulus*	OQ352865	OQ427899	OQ427905	OQ427908	OQ427896	OQ427902	OQ695534	OQ695531
*C. hippeastri* (CBS 241.78)	The Netherlands, *Hippeastrum* sp.	JX010293	JX009740	JX009932	—	JX009485	JX009838	—	—
*C. horii* (ICMP 12942)	New Zealand, *Diospyros kaki*	GQ329687	JX009603	GQ329685	JX010375	JX009533	JX009748	JX010072	—
*C. horii* (**NBRC 7478**)	Japan, *Diospyros kaki*	GQ329690	JX009604	GQ329681	JX010450	JX009438	JX009752	JX010137	—
*C. horii* (ICMP 17968)	China, *Diospyros kaki*	JX010212	JX009605	GQ329682	JX010378	JX009547	JX009811	JX010068	—
*C. musae* (IMI 52264)	Kenya, *Musa sapientum*	JX010142	JX009689	JX010015	JX010395	JX009432	JX009815	JX010084	—
*C. musae* (**CBS 116870**)	USA, *Musa* sp.	JX010146	JX009742	JX010050	HQ596280	JX009433	JX009896	JX010103	KC888926
*C. noveboracense* (AFKH109)	USA, Apple/Idared	MN646685	MN640566	MN640567	MN640569	—	—	MN640568	MN640564
*C. noveboracense* (PMBrms-1)	USA, Apple	MN715324	MN741056	MN741087	MN741064	—	—	MN741100	MN741075
*C. nupharicola* (CBS 469.96)	USA, *Nuphar lutea* subsp. *Polysepala*	JX010189	JX009661	JX009936	JX010397	JX009486	JX009834	JX010087	—
*C. nupharicola* (**CBS 470.96**)	USA, *Nuphar lutea* subsp. *Polysepala*	JX010187	JX009663	JX009972	JX010398	JX009437	JX009835	JX010088	JX145319
*C. queenslandicum* (**ICMP 1778**)	Australia, *Carica papaya*	JX010276	JX009691	JX009934	JX010414	JX009447	JX009899	JX010104	KC888928
*C. queenslandicum* (ICMP 18705)	Fiji, *Coffea* sp.	JX010185	JX009694	JX010036	JX010412	JX009490	JX009890	JX010102	—
*C. salsolae* (**ICMP 19051**)	Hungary, *Salsola tragus*	JX010242	JX009696	JX009916	JX010403	JX009562	JX009863	JX010093	KC888925
*C. salsolae* (CBS 119296)	Hungary, *Glycine max (inoculated)*	JX010241	JX009695	JX009917	—	JX009559	JX009791	—	—
*C. siamense* (**ICMP 18578**)	Thailand, *Coffea arabica*	JX010171	FJ917505	JX009924	JX010404	FJ907423	JX009865	JX010094	—
*C. siamense* (ICMP 17795)	USA, *Malus domestica*	JX010162	JX009703	JX010051	JX010393	JX009506	JX009805	JX010082	—
*C. siamense* (XZEC41 ^c^)	China, *C. betulus*	OQ352866	OQ427927	OQ427933	OQ427936	OQ427924	OQ427930	OQ695549	OQ695547
*C. siamense* (XZEC42)	China, *C. betulus*	OQ352867	OQ427928	OQ427934	OQ427937	OQ427925	OQ427931	OQ695550	OQ695548
*C. theobromicola* (ICMP 17895)	Mexico, *Annona diversifolia*	JX010284	JX009600	JX010057	JX010382	JX009568	JX009828	JX010066	—
*C. theobromicola* (**CBS 124945**)	Panama, *Theobroma cacao*	JX010294	JX009591	JX010006	JX010447	JX009444	JX009869	JX010139	KC790726
*C. tropicale* (MAFF 239933)	Japan, *Litchi chinensis*	JX010275	JX009722	JX010020	JX010396	JX009480	JX009826	JX010086	—
*C. tropicale* (**CBS 124949**)	Panama, *Theobroma cacao*	JX010264	JX009719	JX010007	JX010407	JX009489	JX009870	JX010097	KC790728
*C. xanthorrhoeae* (**BRIP 45094**)	Australia, *Xanthorrhoea preissii*	JX010261	JX009653	JX009927	JX010448	JX009478	JX009823	JX010138	KC790689
*C. xanthorrhoeae* (ICMP 17820)	Australia, *Xanthorrhoea* sp.	JX010260	JX009652	JX010008	—	JX009479	JX009814	—	—

^a^: The number of cultures/isolates in bold represents ex-type strains. ATCC, American Type Culture Collection; BRIP, Plant Pathology Herbarium, Department of Employment, Economic Development and Innovation, Queensland, Australia; CBS, Culture Collection of the Centraalbureau voor Schimmelcultures, Fungal Biodiversity Center, Utrecht, the Netherlands; ICMP, International Collection of Microorganisms from Plants, Auckland, New Zealand; IMI, Culture Collection of CABI Europe UK Centre, Egham, UK; MAFF, MAFF Genebank Project, Ministry of Agriculture, Forestry and Fisheries, Tsukuba, Japan; MFLUCC, Mae Fah Luang University Culture Collection, Chiang Rai, Thailand; NBRC, Biological Resource Center, National Institute of Technology and Evaluation, Japan. *C. hippeastri* (CBS 241.78) was added as an outgroup. ^b^: ITS, internal transcribed spacer gene; CAL, partial calmodulin gene; CHS-1, partial chitin synthase; GAPDH, partial glyceraldehyde 3-phosphate dehydrogenase gene; ACT, partial actin gene; TUB2, partial beta-tubulin 2 gene; ^c^: isolates used for morphological and biological analysis and pathogenicity tests.

## Data Availability

All data generated or analyzed during this study are included in this article.

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
