# Peer review of "Species of the Colletotrichum spp., the Causal Agents of Leaf Spot on European Hornbeam (Carpinus betulus)"

_jof, 2023, doi:10.3390/jof9040489_

Round 1

Reviewer 1 Report

The work reported is interesting. However, there are some points that the authors need to address for publication. The points are given below:

1.       Title: Collectotrichum gloeosprioides should change Collectotrichum spp. although 4 species were mentioned in lines 58-60

2.       A brief description of methods used in the study, the aim of the study, and the source of the isolates should be placed in the abstract. Thus, this abstract should be rewritten.

3.       In the keywords, the names of the fungi and the plant were incorrectly presented, they should be rewritten.

4.       The infected leaves had been collected throughout 10 months, which is a fairly long time. The methods to store the leaf samples or the isolated fungi should be introduced. Moreover, when were the fungi isolated from the sampled leaves? Were the leaves analyzed at the same time? Or were the fungi isolated separately? This should be clarified.

5.       The location of the study, along with its natural features, needed to be shown in the materials and method. Moreover, the source of the European hornbeam should be detailed, instead of “a commercial orchard”.

6.       Having had five replicates, why were the experiment conducted twice?

7.       Numbers above 999, should have a coma separating every 3 digits, e.g. 1,000,000.

8.       Line 148-149, a total number of trees is required. Twenty percent of the total trees had symptoms, and only 15-20% of “the trees” had their leaves observed. What is the meaning of “the tree”? Is it the total trees or only the trees with symptoms? Moreover, why were only 15-20% leaves observed instead of all of them?

Line 148-149 should move to introduction or discussion

9.       In the Figure 1, the distance between trees was too close, cross infection may occur and there is a lack of experimental design, making unreliable data.

10.   Based on what morphology of colonies were the strains of Colletotrichum divided? This should be explained in the results and it is needed to have pictures of the colonies from the 23 isolates. And why were the Fusarium studied, while they may affect the disease?

11.   The term of random selection was not suitable, because each isolate may differ at certain levels. The isolates should not have been selected randomly for further research, they should have applied equally. This can reduce the reliability of the study.

12.   In the pathogenicity test, a basis of virulence rate is required, judging the symptoms by naked eyes is unreliable. Therefore, a proper measurement and ladders for the symptoms are needed.

13.   The study of the isolation of C. aenigma in 2020 should be cited.

14.   The discussion needs to mention about the disease caused by the Colletotrichum.

15.   The conclusion should be separated from the discussion. How about conclusion section

Author Response

Dear Reviewer: 

Thank you for your comments concerning our manuscript entitled “Species of the Colletotrichum gloeosprioides species complex, the causal agents of leaf spot on European hornbeam (Carpinus betulus)” (ID= jof-2278521). Those comments are all valuable and very helpful in revision and improvement, which is of great guiding significance to our research. We have studied comments carefully and have made corrections which we hope to meet with approval. Revised portion are marked in red in the paper, please see the attachment. The main corrections in paper and the responses are as following:

Q1: Title: Collectotrichum gloeosprioides should change Collectotrichum spp. although 4 species were mentioned in lines 58-60

Response 1: Thanks for your comment. We have revised in line 2.

Q2: A brief description of methods used in the study, the aim of the study, and the source of the isolates should be placed in the abstract. Thus, this abstract should be rewritten.

Response 2: Thanks for your comment. We have rewritten the abstract in line 10-22.

Q3: In the keywords, the names of the fungi and the plant were incorrectly presented, they should be rewritten.

Response 3: Thanks for your comment. We have revised in line 23.

Q4: The infected leaves had been collected throughout 10 months, which is a fairly long time. The methods to store the leaf samples or the isolated fungi should be introduced. Moreover, when were the fungi isolated from the sampled leaves? Were the leaves analyzed at the same time? Or were the fungi isolated separately? This should be clarified.

Response 4: Thanks for your comment. We isolated the fungi in the second day after field survey. We have revised in line 122. The method to isolated fungi were revised in line122-135. During the period that between two field survey, the fungi were stored at 4℃. The fungi were isolated separately.

Q5: The location of the study, along with its natural features, needed to be shown in the materials and method. Moreover, the source of the European hornbeam should be detailed, instead of “a commercial orchard”.

Response 5: Thank you for your advice. We have revised in line 116-119 and 137-138.

Q6: Having had five replicates, why were the experiment conducted twice?

Response 6: We conducted the experiment twice just want to verify the pathogenicity in second time.

Q7: Numbers above 999, should have a coma separating every 3 digits, e.g. 1,000,000.

Response 7: Thank you for your advice. We have added the punctuation every 3 digits of the numbers throughout this manuscript.

Q8: Line 148-149, a total number of trees is required. Twenty percent of the total trees had symptoms, and only 15-20% of “the trees” had their leaves observed. What is the meaning of “the tree”? Is it the total trees or only the trees with symptoms? Moreover, why were only 15-20% leaves observed instead of all of them?

Response 8: Sorry, “the tree” means the diseased European hornbeam, and the 15-20% leaves were the percentage of diseased laves not the total observation. We have revised in line 206-210.

Q9: In the Figure 1, the distance between trees was too close, cross infection may occur and there is a lack of experimental design, making unreliable data.

Response 9: The European hornbeam in Figure 1 were the diseased trees in Xuzhou city, not the trees that we conducted pathogenicity test.

Q10: Based on what morphology of colonies were the strains of Colletotrichum divided? This should be explained in the results and it is needed to have pictures of the colonies from the 23 isolates. And why were the Fusarium studied, while they may affect the disease?

Response 10: Sorry, we divide was according to the density of hyphae and the distribution of pigment on the reverse side of colonies, it was obvious as we described in 3.3 morphological characteristics, so we did not take photos at that time, and we have revised in line 228-229. There have been reports of Fusarium causing leaf spot disease before, so we made a preliminary study. But we found that it was not pathogenic, and because we changed the positions of 3.3 and 3.4, so we have deleted the content about Fusarium for the sake of logical fluency.

Q11: The term of random selection was not suitable, because each isolate may differ at certain levels. The isolates should not have been selected randomly for further research, they should have applied equally. This can reduce the reliability of the study.

Response 11: Thank you for your advice, the isolates we selected was based on its representative colony morphology, we have revised the term of selection in line 232, 327, and 351.

Q12: In the pathogenicity test, a basis of virulence rate is required, judging the symptoms by naked eyes is unreliable. Therefore, a proper measurement and ladders for the symptoms are needed.

Response 12: Thank you for your advice, we just want to describe the distribution and size of the lesions, we will pay attention to this question in future experiments. We have revised in line 333-347.

Q13: The study of the isolation of C. aenigma in 2020 should be cited.

Response 13: Thank you for your advice, we have cited it in line 488.

Q14: The discussion needs to mention about the disease caused by the Colletotrichum.

Response 14: Thank you for your advice, we have revised in line 525-532.

Q15: The conclusion should be separated from the discussion. How about conclusion section

Response 15: Thank you for your advice, we have separated conclusion section from discussion in line 541-544.

Sincerely yours,

Corresponding Author:

Prof. Chen

Nanjing Forestry University

College of Forestry

E-Mail: cfengmao@njfu.edu.cn

Reviewer 2 Report

The manuscript reports results of a study on European hornbeam - the first report of Colletotrichum species causing leaf spot on this species of tree. In general, the manuscript is well and clear written, however, I have made several comments especially on the methods part, since the authors give too little information in few paragraphs. Furthermore, I would suggest to include more gene loci for the phylogenetic tree (please see detailed comment below). The conclusion is not suitable in my opinion, see specific comment below.

Detailed Comments

Line 28 worldwide?

Line 36 is it actually used to combat cancer, if yes, which one?

Line 53-70 please see recently published paper of Astolfi et al 2022 and include missing information, such as gene loci and species (DOI: 10.1094/PHYTO-12-21-0527-SC)

Line 82 pathogenic bacteria? I assume that you mean fungi?

Line 88-89 where? In one or more commercially used fields or private garden/s? How was the sampling done? Randomly? Which part of tree? How much samples from how much trees? Please be more specific.

Line 97 Please specify the cleaning of the leaves …

Line 98 condia/ml à conidia ml-1

Which device have you used for spraying and how much of spore suspension have you sprayed on one tree? Have you used the whole tree for the inoculation or only parts/branches/leaves?

Line 121ff The taxonomy of Colletotrichum sp. is a constant challenge and further loci, such as Glutamine Synthetase, DNA Lyase and the Intergenic spacer and partial mating type (Mat 1-2) gene were needed to distinguish between individual clades of the CGSC. Recently Khodadadi et al. 2020; Martin et al. 2021 described the new species C. chrysophilum and C. noveboracense, which are closely related to C. fructicola. Furthermore C. chrysophilum was recently identified to cause Glomerella leaf spot on apples, although C. fructicola was first identified as causal agent (see Astolfi et al). Therefore I would highly suggest to include the above mentioned gene loci in your phylogenetic analysis.

Line 147ff I still miss a detailed description of the field, as already mentioned above…

Line 154 delete the last sentence

Line 158-163 can you please refer to the foto of the different (4 groups) fungal colonies?

Line 171-175 How have you analyzed/measured the virulence of the isolates and how was the classification in weak, moderate and strong done?

Line 179 I would suggest moving the paragraph 3.4 Morphological characteristics above the paragraph 3.3 Pathogenicity test

Line 165 until here you have grouped the isolates morphological. So I wonder how you can write that you have 4 different species, since it is impossible to characterize the isolates morphological. Same applies to line 180. Think about to change the order of the paragraphs!!!

Line 242 Please include C. chrysophilum and C. noveboracense in the phylogenetic tree

Line 244 Why have you used only 2 strains from group 4 and 3 from all other groups?

Line 292ff what deviations are we talking about?

Line 306 please write this part more positively, yes, it is confusing but it is the best indication, that it is not possible to identify the Colletotrichum species morphological, even if it is 100 % the same secies, likely due to other incubation conditions (light, temperature, humidity, position in the incubator….)

Line 324ff it is a bit a strange end of the whole story, I think I understand that you try to give a practical hint to the whole story, but I would suggest rewriting this part and include also the strength of this paper, the importance of your analysis – combine it with the practical relevance and ideas for future studies and come back to the introduction

Author Response

Dear Ms. Yasmine Wang: 

Thank you for your comments concerning our manuscript entitled “Species of the Colletotrichum gloeosprioides species complex, the causal agents of leaf spot on European hornbeam (Carpinus betulus)” (ID= jof-2278521). Those comments are all valuable and very helpful in revision and improvement, which is of great guiding significance to our research. We have studied comments carefully and have made corrections which we hope to meet with approval. Revised portion are marked in red in the paper, please see the attachment. The main corrections in paper and the responses are as following:

Q1: Line 28 worldwide?

Response 1: Sorry, the applications mentioned are mainly in Iran, and we have revised in line 29

Q2: Line 36 is it actually used to combat cancer, if yes, which one?

Response 2: Sorry, the cancer was not specifically mentioned in the literature, and the experiments they conducted was growth inhibition test in vitro.

Q3: Line 53-70 please see recently published paper of Astolfi et al 2022 and include missing information, such as gene loci and species (DOI: 10.1094/PHYTO-12-21-0527-SC)

Response 3: Thank you for your advice, we have added the GS and ApMat genes in line 84-85.

Q4: Line 82 pathogenic bacteria? I assume that you mean fungi?

Response 4: We are very sorry for our negligence. We have revised in line 110.

Q5: Line 88-89 where? In one or more commercially used fields or private garden/s? How was the sampling done? Randomly? Which part of tree? How much samples from how much trees? Please be more specific. 

Response 5: Thank you for your advice. We have added the specific information in line 116-122.

Q6: Line 97 Please specify the cleaning of the leaves …

Response 6: Thank you for your advice. We have added the cleaning method in line 139-141

Q7: Line 98 condia/ml à conidia ml-1

Response 7: Thank you for your advice, we have revised in line 142.

Q8: Which device have you used for spraying and how much of spore suspension have you sprayed on one tree? Have you used the whole tree for the inoculation or only parts/branches/leaves?

Response 8: We sprayed with a 10 ml plastic sprinkling can and we sprayed approximately 2-3ml on one tree. We used only one branch of leaves for inoculation, and we cover a plastic bag when spraying and the plastic bag would reserve for 1-2 days to prevent the conidia from escaping.  

Q9: Line 121ff The taxonomy of Colletotrichum sp. is a constant challenge and further loci, such as Glutamine Synthetase, DNA Lyase and the Intergenic spacer and partial mating type (Mat 1-2) gene were needed to distinguish between individual clades of the CGSC. Recently Khodadadi et al. 2020; Martin et al. 2021 described the new species C. chrysophilum and C. noveboracense, which are closely related to C. fructicola. Furthermore C. chrysophilum was recently identified to cause Glomerella leaf spot on apples, although C. fructicola was first identified as causal agent (see Astolfi et al). Therefore, I would highly suggest to include the above mentioned gene loci in your phylogenetic analysis.

Response 9: Thank you for your advice. As described by Liu et al. 2022, ApMat, CAL, and GS were the specific loci used for CGSC, APN2 was used for C. caudatum and C. graminicola species complexes, and four Colletotrichum species in this study are belonging to CGSC, so we decide to add ApMat and GS loci in our study. And we have added the species of C. chrysophilum and C. noveboracense.

Q10: Line 147ff I still miss a detailed description of the field, as already mentioned above

Response 10: We are sorry about that. We have added the description in line 206-210

Q11: Line 154 delete the last sentence

Response 11: Thank you for your advice, we have revised in line 215

Q12: Line 158-163 can you please refer to the foto of the different (4 groups) fungal colonies?

Response 12: Sorry, we divide was according to the density of hyphae and the distribution of pigment on the reverse side of colonies, it was obvious as we described in 3.3 morphological characteristics, so we did not take photos at that time.

Q13: Line 171-175 How have you analyzed/measured the virulence of the isolates and how was the classification in weak, moderate and strong done?

Response 13: Thank you for your patient comment. We just want to describe the distribution and size of the lesions. We will pay attention to this question in future experiments. We have revised in line 333-337.

Q14: Line 179 I would suggest moving the paragraph 3.4 Morphological characteristics above the paragraph 3.3 Pathogenicity test

Response 14: Thank you for your advice, we have changed the order of the paragraphs

Q15: Line 165 until here you have grouped the isolates morphological. So I wonder how you can write that you have 4 different species, since it is impossible to characterize the isolates morphological. Same applies to line 180. Think about to change the order of the paragraphs!!!

Response 15: Sorry, this is our negligence that we used the wrong word. We originally intended to use ‘group’ to describe these four categories of Colletotrichum instead of misusing the word ‘species’. We have changed the order of the paragraphs and we have revised ‘species’ into ‘group’.

Q16: Line 242 Please include C. chrysophilum and C. noveboracense in the phylogenetic tree

Response 16: Thank you for your advice, we have added C. chrysophilum and C. noveboracense in the phylogenetic tree.

Q17: Line 244 Why have you used only 2 strains from group 4 and 3 from all other groups?

Response 17: Considering that the morphology characteristics of within each Colletotrichum group were almost same, so we selected 3 representative strains from group 1-3, and the group 4 only has 2 strains in total, so we selected them all, and the quantity of each group was mentioned in line 229-230.

Q18: Line 292ff what deviations are we talking about?

Response 18: We are talking about the differences of morphology between four Colletotrichum species in this study and previous studies. We have added the information in line 474-475.

Q19: Line 306 please write this part more positively, yes, it is confusing but it is the best indication, that it is not possible to identify the Colletotrichum species morphological, even if it is 100 % the same species, likely due to other incubation conditions (light, temperature, humidity, position in the incubator….)

Response 19: Thank you for your advice, we have revised this part in line 500-508.

Q20: Line 324ff it is a bit a strange end of the whole story, I think I understand that you try to give a practical hint to the whole story, but I would suggest rewriting this part and include also the strength of this paper, the importance of your analysis – combine it with the practical relevance and ideas for future studies and come back to the introduction

Response 20: Thank you for your advice, we have rewritten the end in line 533-539.

Sincerely yours,

Corresponding Author:

Prof. Chen

Nanjing Forestry University

College of Forestry

E-Mail: cfengmao@njfu.edu.cn

Round 2

Reviewer 2 Report

The manuscript has improved, however I have still few comments….please check English spelling throughout the new parts of the manuscript

Line 59: I don’t like the words “and so on…” please delete them

Line 90-96 as already mentioned before, be more specific… 30 diseased samples were sampled and pooled or have you kept them separately? Have you used 30 isolate…I don`t get it… What do you mean with abundant sunshine, moderate rainfall…?? Do you have any numbers or can describe it more specifically? Is this the only garden with infected trees or do you have more info about the distribution?

Line 104-108 please check the spelling!!!!! Have you used all leaves of the saplings for the inoculation? Please be more specific. Please include response 8 in the manuscript

Response 13 I can’t find the right part/line with this line specification???

Conclusion? I’don’t get it.. you have now 2 conclusions in the manuscript… please combine them, since there are many overlaps

Author Response

Dear Ms. Yasmine Wang: 

Thank you for your comments concerning our manuscript entitled “Species of the Colletotrichum gloeosprioides species complex, the causal agents of leaf spot on European hornbeam (Carpinus betulus)” (ID= jof-2278521). Those comments are all valuable and very helpful in revision and improvement, which is of great guiding significance to our research. We have studied comments carefully and have made corrections which we hope to meet with approval. Revised portion are marked in red in the paper, please see the attachment. The main corrections in paper and the responses are as following:

Q1: Line 59: I don’t like the words “and so on…” please delete them

Response 1: Thank you for your advice, we have replaced the words “and so on…” into “etc.”

Q2: Line 90-96 as already mentioned before, be more specific… 30 diseased samples were sampled and pooled or have you kept them separately? Have you used 30 isolates…I don`t get it… What do you mean with abundant sunshine, moderate rainfall…?? Do you have any numbers or can describe it more specifically? Is this the only garden with infected trees or do you have more info about the distribution?

Response 2: We keep diseased samples separately in different sampling bag separately. We did not use 30 isolates, 28 strains were isolated from the margin pieces in total, five of them were Fusarium and they were non-pathogenic, so there are 23 strains used in this study. “Abundant sunshine, moderate rainfall” means, the annual sunshine hours are 2284-2495 hours, the sunshine rate is 52%-57%, the average annual precipitation is 800-930 mm, and the rainy season precipitation accounts for 56% of the whole year. We have revised in line 92-94.And sorry, we only investigated the garden, but not the surrounding area.

Q3: Line 104-108 please check the spelling!!!!! Have you used all leaves of the saplings for the inoculation? Please be more specific. Please include response 8 in the manuscript.

Response 3: Thank you for your advice. We have added response 8 in line 113-114. We did not used all leaves, each treatment contained five leaves and we have revised in line 115.

Q4: Response 13 I can’t find the right part/line with this line specification???

Response 4: Sorry, the right line is 259-263.

Q5: Conclusion? I don’t get it. you have now 2 conclusions in the manuscript… please combine them, since there are many overlaps 

Response 5: Thank you for your advice. We have deleted overlaps and revised in line 399-405.

Sincerely yours,

Corresponding Author:

Prof. Chen

Nanjing Forestry University

College of Forestry

E-Mail: cfengmao@njfu.edu.cn